# Development and Validation of a Novel Pre-Pregnancy Score Predictive of Preterm Birth in Nulliparous Women Using Data from Italian Healthcare Utilization Databases

**DOI:** 10.3390/healthcare10081443

**Published:** 2022-08-01

**Authors:** Ivan Merlo, Anna Cantarutti, Alessandra Allotta, Elisa Eleonora Tavormina, Marica Iommi, Marco Pompili, Federico Rea, Antonella Agodi, Anna Locatelli, Rinaldo Zanini, Flavia Carle, Sebastiano Pollina Addario, Salvatore Scondotto, Giovanni Corrao

**Affiliations:** 1Department of Statistics and Quantitative Methods, University of Milano-Bicocca, 20126 Milan, Italy; ivan.merlo@unimib.it (I.M.); federico.rea@unimib.it (F.R.); giovanni.corrao@unimib.it (G.C.); 2National Centre for Healthcare Research and Pharmacoepidemiology, University of Milano-Bicocca, 20126 Milan, Italy; f.carle@univpm.it (F.C.); walter.pollina.ext@regione.sicilia.it (S.P.A.); salvatore.scondotto@gmail.com (S.S.); 3Department of Health Activities and Epidemiological Observatory, Regional Health Authority, Sicily Region, 90145 Palermo, Italy; aleallotta@gmail.com (A.A.); elisaeleonora.tavormina@regione.sicilia.it (E.E.T.); 4National Research Council of Italy, Institute for Biomedical Research and Innovation, 90146 Palermo, Italy; 5Center of Epidemiology, Biostatistics and Medical Information Technology, Department of Biomedical Sciences and Public Health, Marche Polytechnic University, 60020 Ancona, Italy; m.iommi@staff.univpm.it; 6Regional Epidemiological Observatory, Regional Health Agency of Marche, 60125 Ancona, Italy; marco.pompili@regione.marche.it; 7Department of Medical and Surgical Sciences and Advanced Technologies “GF Ingrassia”, University of Catania, 95123 Catania, Italy; antonella.agodi@unict.it; 8Department of Mother and Child, ASST Vimercate, 20871 Vimercate, Italy; anna.locatelli@unimib.it; 9School of Medicine and Surgery, University of Milano Bicocca, 20900 Monza, Italy; 10Past Director of Woman and Child Health Department, Azienda Ospedaliera della Provincia di Lecco, 23900 Lecco, Italy; rinaldozanini@gmail.com; 11Directorate General for Health, Lombardy Region, 20124 Milan, Italy

**Keywords:** preterm birth, score, real-world evidence, healthcare utilization database, nulliparous

## Abstract

*Background:* Preterm birth is a major worldwide public health concern, being the leading cause of infant mortality. Understanding of risk factors remains limited, and early identification of women at high risk of preterm birth is an open challenge. *Objective:* The aim of the study was to develop and validate a novel pre-pregnancy score for preterm delivery in nulliparous women using information from Italian healthcare utilization databases. *Study Design:* Twenty-six variables independently able to predict preterm delivery were selected, using a LASSO logistic regression, from a large number of features collected in the 4 years prior to conception, related to clinical history and socio-demographic characteristics of 126,839 nulliparous women from Lombardy region who gave birth between 2012 and 2017. A weight proportional to the coefficient estimated by the model was assigned to each of the selected variables, which contributed to the Preterm Birth Score. Discrimination and calibration of the Preterm Birth Score were assessed using an internal validation set (i.e., other 54,359 deliveries from Lombardy) and two external validation sets (i.e., 14,703 and 62,131 deliveries from Marche and Sicily, respectively). *Results:* The occurrence of preterm delivery increased with increasing the Preterm Birth Score value in all regions in the study. Almost ideal calibration plots were obtained for the internal validation set and Marche, while expected and observed probabilities differed slightly in Sicily for high Preterm Birth Score values. The area under the receiver operating characteristic curve was 60%, 61% and 56% for the internal validation set, Marche and Sicily, respectively. *Conclusions:* Despite the limited discriminatory power, the Preterm Birth Score is able to stratify women according to their risk of preterm birth, allowing the early identification of mothers who are more likely to have a preterm delivery.

## 1. Introduction

Preterm birth (PTB), defined as any live birth occurring before 37 completed weeks of gestation [1], is the leading cause of neonatal mortality, and remains the most common cause of death under 5 years of age, with over 1 million children dying annually worldwide [2]. PTB also increases the newborn’s risk of dying due to other causes, mainly from neonatal infections [3], and has long-term medical and social consequences, including an increased risk of altered cardiovascular and renal function, cerebral palsy, mental retardation and disability [4,5].

The rate of PTB has increased over the past decades, reaching the value of 11.1% worldwide, ranging from 9.3% in high-income countries to 11.8% in low-income ones [6]. Several maternal characteristics have been associated with PTB, including gestational and pre-gestational factors. Gestational risk factors include poor nutritional status, stress, substance abuse (smoke, alcohol, drugs), multiple gestation, intra and extra-uterine infections and inflammations [7]. In the past decades, new gestational markers have emerged. Low cervical length and biomarkers, such as fetal fibronectin, as well as maternal hypertensive pathology and fetal underdevelopment, have shown an association with PTB, although predictive accuracy remains limited [8]. Pre-gestational risk factors include the use of assisted medical conception techniques [9], ethnic and socio-demographic characteristics, maternal age, medical disorders and history of preterm delivery, which is considered one of the strongest risk factors, with recurrence rates ranging from 15% to 50% depending on the number and gestational age of previous deliveries [7].

While many studies have evaluated the association between risk factors and preterm delivery, a large proportion of PTBs remains unexplained. A 2016 multinational study involving 4.1 million births of five countries reported that approximately two out of three PTBs lack a plausible biological explanation [10].

Better identification of women at high risk of PTB, already at the beginning of the pregnancy, is a desirable goal as it could lead to a reduction in the rate of preterm delivery by facilitating monitoring and timely intervention. Such an assessment is particularly useful in nulliparous women. Furthermore, being able to predict preterm delivery could be useful to better understand the mechanisms leading to PTB [11]. To date, researchers have mainly focused on studying specific pre-gestational conditions. However, the widespread adoption of electronic health databases now offers the opportunity to consider a large part of the mother’s medical history when studying the association between maternal characteristics and preterm delivery, without making *a priori* assumptions about the predictors of PTB.

The aim of the study was to develop and validate a predictive pre-pregnancy score for preterm delivery in nulliparous women using the information collected in healthcare utilization databases of the Italian National Health Service (NHS) during the four years prior to conception.

## 2. Methods

### 2.1. Setting

The present study was based on the NHS beneficiaries of three Italian regions that joined the protocol and contributed to the data collection. The regions are located in Northern (Lombardy), Central (Marche) and Southern (Sicily) Italy, covering approximately 16.3 million people (almost 30% of the Italian population).

### 2.2. Data Sources

All Italian citizens have equal access to healthcare services as part of the NHS. Within each of the 21 Italian regions and autonomous provinces, computerized information systems have been created to collect a variety of data regarding beneficiaries who receive NHS assistance and the provided services. Collected information includes: (i) demographic and administrative data of beneficiaries of the NHS; (ii) hospital discharge records reporting information on diagnoses and procedures received during hospitalization, coded according to the International Classification of Diseases, 9th Revision, Clinical Modification (ICD9-CM); (iii) drug prescriptions reimbursed by the NHS, coded according to the Anatomical Therapeutic Chemical (ATC) classification system; (iv) records on services provided on an outpatient basis (e.g., outpatient visits, diagnostic exams); (v) health exemptions granted to citizens, identified by a specific national code; (vi) data from Certificates of Delivery Assistance (CedAP) including information self-reported by the mother relating to her socio-economic traits, other medical information relating to pregnancy, childbirth and child health status at delivery. These various types of data are linked using, for each citizen, a single identification code recorded in all databases. To preserve privacy, each identification code is automatically deidentified. Analyses of the regional databases were performed under the rule that the inverse process, that is, patient identification, was allowed only to the Regional Health Authority upon request from the judicial authority.

### 2.3. Score Development

Since Lombardy has the largest resident population (16% of the entire Italian population), data from this region were used to develop the score. Deliveries occurred from January 2012 to December 2017 were selected through the CedAP registry. Only women with age at delivery between 15 and 55, gestational age between 22 and 42 weeks, and at least 4 years of traceability in healthcare databases before pregnancy were included. Records that lacked important information about the mother or child, as well as incorrect records (i.e., duplicates or records of women for whom no hospital admission reporting an ICD-9-CM code for delivery was found), and deliveries that resulted in no babies born alive were excluded. Nulliparous women were selected combining information from CedAP registry and hospital discharge database. For each woman, the 4 years preceding the beginning of gestation were taken into account. The available information made it possible to outline the profile of women during that period with respect to (i) drugs intake, (ii) health exemptions coverage, (iii) diagnoses and (iv) procedures received during hospitalization, (v) outpatient services used, and (vi) socio-demographic features. Drugs were grouped into categories based on the second level of the ATC code (i.e., first three digits). With regard to exemptions due to chronic diseases, the entire national code was taken into account, while exemptions for disabilities and other socio-economic conditions were grouped according to the first digit of the code. Diagnoses and procedures received during hospitalization were grouped considering the first three digit of the ICD9-CM code. Outpatient services were distinguished according to the code of the medical branch to which they belonged. Each woman was considered exposed to a certain factor (i.e., a variable among those listed above) if the factor code was reported at least once in the healthcare utilization databases during the considered time. Socio-demographic traits were extracted from the CedAP registry and included: (i) age at conception (≤25, 26–35 and ≥36 years); (ii) employment status, categorized as unemployed and employed (the latter including both working women and students); (iii) marital status (married or unmarried); (iv) education, measured according to the length of formal education completed and categorized as ≤12 (low), from 13 to 15 (intermediate), and ≥16 years (high); (v) country of birth (Italy or abroad). Moreover, for each woman the type of conception was taken into account; a binary variable that indicated the use or not of assisted medical conception techniques was assigned to each subject, in accordance with what reported in the CedAP. Deliveries resulting in PTB were identified through the same database.

A training set containing 70% of the Lombardy cohort was randomly selected and used to develop the score. With the aim of selecting variables independently able to predict PTB, the least absolute shrinkage and selection operator (LASSO) method was applied, via logistic regression, on the training set. LASSO selects variables correlated to the outcome by shrinking coefficient values, down to zero for the ones not correlated to outcome [12]. The LASSO tuning parameter was chose using a 10-fold cross-validation, which allowed selecting the parameter that guaranteed the best discriminant power for the model [13]. Only the variables that had an absolute frequency greater than or equal to 5 in the training set were considered as candidate predictors in the selection process. The absolute frequency of 5 was arbitrary chosen as a compromise between: (i) the need to reduce the number of candidate predictors in order to have acceptable computational times (this criterion made it possible to reduce the number of covariates by 40%, see Results); (ii) avoid the selection of noise variables; and (iii) limit the exclusion of conditions associated with PTB that could positively impact the score performances. The coefficients estimated by the LASSO model were then used to assign an integer weight to each selected condition. In particular, the weights were calculated proportionally to the value of the respective coefficients, making sure the sum of all weights was equal to 100. For each woman, a total aggregate score was identified by sequentially summing the weights of the conditions to which she was exposed. To simplify the system, accounting for excessive heterogeneity of the total aggregate score, the latter was categorized by assigning increasing values of 0, 1, 2 and 3 to the categories of 0–2, 3–6, 7–15 and 16–100, respectively. The index so obtained was termed Preterm Birth Score (PTBS).

Codes used to identify the factors included in the score are shown in the Appendix A.

The predictive performance of the score was initially evaluated in the training set by constructing the receiver operating characteristic (ROC) curve and calculating the corresponding underlying area (area under the ROC curve (AUC)).

### 2.4. Score Validation

To assess the reproducibility and generalizability of the results, PTBS was validated under different scenarios, varying temporal and geographical conditions [14]. Firstly, an internal validation procedure was carried out considering the 30% of the cohort from Lombardy region that was excluded in the score development phase. Subsequently, the model was externally validated by considering two different cohorts, respectively, from Marche and Sicily, selected using the same inclusion/exclusion criteria as the original one except for the recruitment period (2015–2019 and 2016–2019 for Marche and Sicily, respectively).

For each validation cohort, the predictive performance was assessed through discrimination and calibration. Discrimination was evaluated by the ROC curves and the corresponding AUCs. Calibration plots displayed observed versus predicted PTB probabilities. With the aim of taking into account the different incidences of the outcome in the different regions, calibration plots were adjusted via a conservative model recalibration by updating the model intercept [15].

### 2.5. Statistical Software

The variables’ selection procedure was performed in R (version 3.5.3) using the ‘glmnet’ package [16]. All other analyses were performed using the Statistical Analysis System Software (version 9.4; SAS Institute, Cary, NC, USA).

## 3. Results

### 3.1. PTBS Development

The 486,400 deliveries that took place in Lombardy between 2012 and 2017 were selected. We sequentially excluded (i) 26 records with maternal age at delivery below 15 or above 55 years, (ii) 489 records with gestational age below 22 or above 42 weeks, (iii) 89,689 deliveries from women without at least 4 years of observation in regional databases before the onset of pregnancy, (iv) 14,186 records that lacked information about the mother, (v) 1594 deliveries from women with no hospital admission, (vi) 1924 records with incorrect linkage or resulted in no babies born alive and (vii) 197,294 deliveries from not nulliparous women (Appendix A). Of the 181,198 remaining records, 126,839 (70%) were randomly selected and used as the training set. In the latter, the proportion of preterm births was equal to 7.4%.

A total of 1771 candidate predictors were identified. Among these, 1056 had an absolute frequency of at least 5 and were considered as candidate predictors in the LASSO selection process. This set of variables included (i) 60 drugs, (ii) 482 hospital diagnosis, (iii) 422 inpatient procedures, (iv) 44 exemptions, (v) 42 outpatient services, (vi) 5 socio-demographic conditions and (vii) the variable that identifies the use of assisted medical conception techniques.

Twenty-six variables were selected and included in the PTBS. Weights and frequencies of the conditions are shown in Table 1, while estimates of the regression coefficients are shown in Appendix A. Factors most associated with PTB (i.e., factors with weight ≥8) were inpatient procedures for other operations on rectum and perirectal tissue, intake of pancreatic hormones, use of assisted medical conception techniques, hospitalization diagnosed with heart failure and presence of an exemption for transplant recipients. Considering their frequency, the variables that contributed the most to the total aggregate score in the training set were intake of sex hormones and modulators of the genital system (weight 4; frequency 20.4%), use of assisted medical conception techniques (weight 10; frequency 6.0%) and age at conception ≥36 years (weight 3; frequency 19.6%).

As an example of PTBS calculation, suppose a woman used assisted medical conception techniques (weight = 10), was more than 36 years old at conception (weight = 3) and registered an exemption for affections of the circulatory system (weight = 1) within the previous four years. Her total aggregate score would be 14 and the corresponding PTBS value would be equal to 2.

Overall, 65.2% and 3.7% of the training set women, respectively, had the lowest (0) and the highest (3) PTBS value, and the AUC value was equal to 0.61 (95% CI: 0.60–0.61) (Figure 1). Probability of preterm delivery was 5.5%, 7.9%, 14.2% and 23.1% for PTBS value equal to 0, 1, 2 and 3, respectively.

### 3.2. PTBS Validation

Preterm birth frequency was equal to 7.2%, 6.2% and 5.7% in the internal validation set (Lombardy), Marche and Sicily, respectively. PTBS distribution in the validation sets was very similar to that observed in the training set, and the AUC value was equal to 0.60 (95% CI: 0.59–0.61), 0.61 (95% CI: 0.59–0.62) and 0.56 (95% CI: 0.55–0.57) in the internal validation set, Marche and Sicily, respectively (Figure 2).

Calibration plots showed that observed preterm birth probabilities for PTBS values in the three validation sets were almost identical to those expected from training set results except for the highest PTBS value in Sicily region, which had observed risk lower than expected, but still higher than the risk observed in lower levels (Figure 3). The interpolation of the calibration curves reflected this situation. Calibration intercept was equal to the ideal value of 0 in both the internal validation set and Marche region, while it was equal to 0.02 in Sicily. Similarly, calibration slope was close to ideal value of 1 in both the internal validation set (0.95) and Marche region (0.96), while it was lower in Sicily (0.67).

**Figure 1 healthcare-10-01443-f001:**
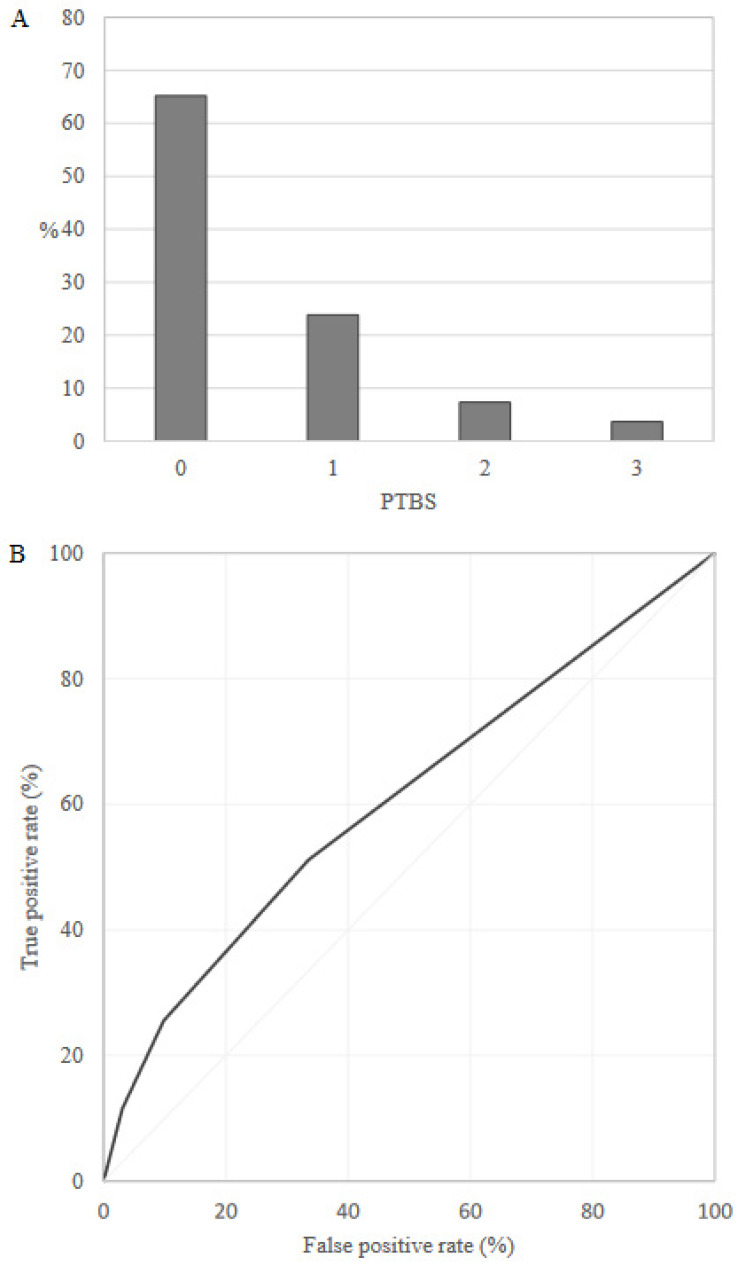
Preterm Birth Score (PTBS) distribution (**A**) and receiver operating characteristic (ROC) curve (**B**) in the training set (Lombardy region).

**Figure 2 healthcare-10-01443-f002:**
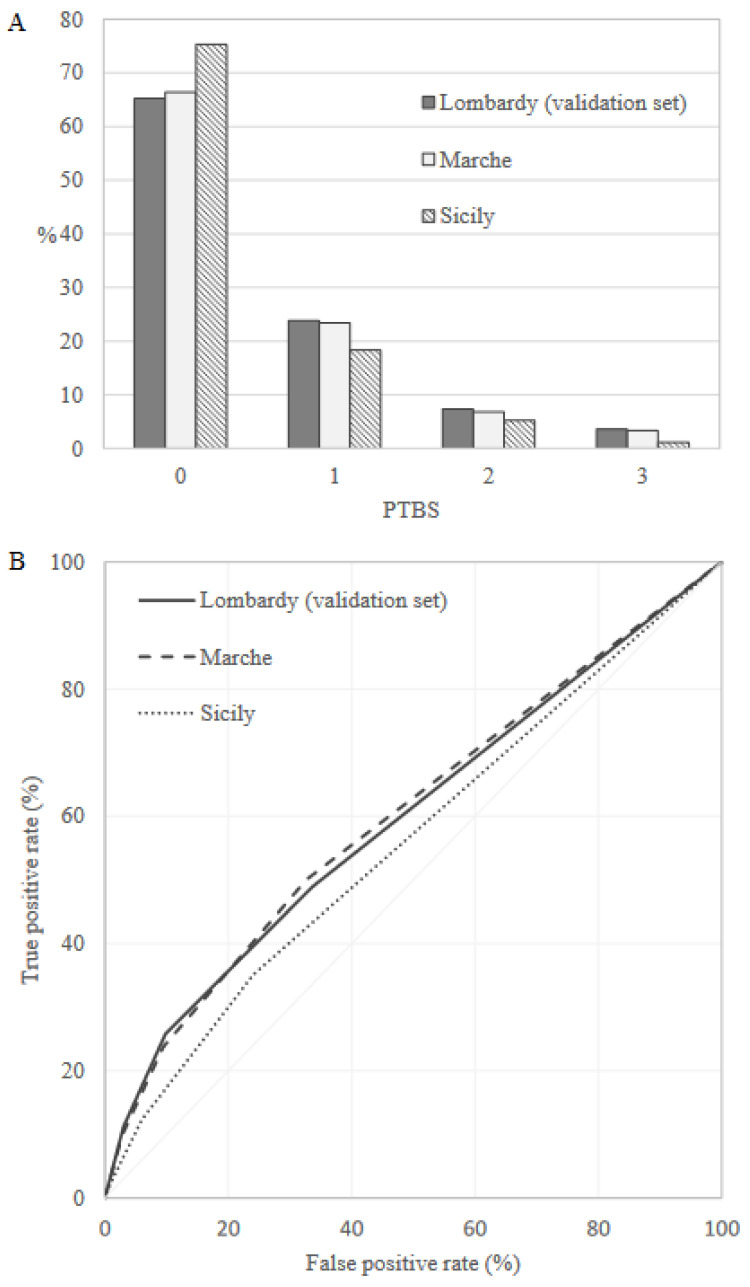
Preterm Birth Score (PTBS) distributions (**A**) and receiver operating characteristic (ROC) curves (**B**) comparing discriminant power in the validation sets.

**Figure 3 healthcare-10-01443-f003:**
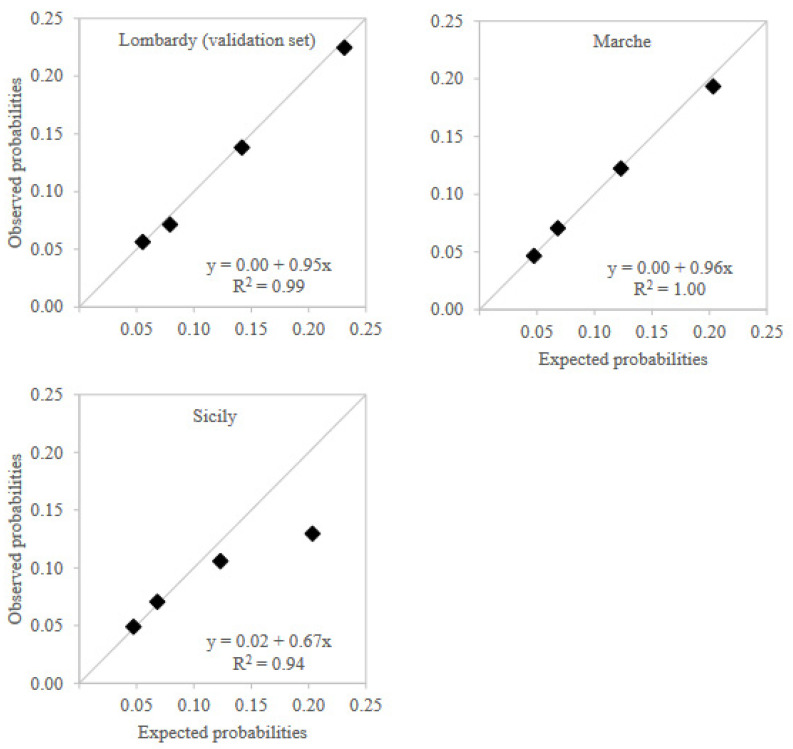
Calibration plots across validation territories.

Notes: Expected probabilities for Marche and Sicily regions were adjusted via a conservative model recalibration by updating the model intercept to take into account the different incidences of preterm birth between regions.

Descriptive statistics of factors included in the PTBS for validation sets were reported in the Appendix A.

## 4. Comment

### 4.1. Principal Findings

In this study, a new predictive score for preterm delivery was developed using the Italian healthcare utilization databases, considering both the woman’s medical history and her socio-demographic conditions. Only nulliparous women were considered and the variables contributing to the score were selected from a large number of candidate predictors already available at the time of conception. These characteristics make PTBS a tool applicable to every woman from the very beginning of pregnancy. Furthermore, the absence of *a priori* hypotheses on the nature of predictors is an innovative approach to the topic and represents an element of novelty in the present work.

Albeit PTBS showed limited discriminatory power, the score is able to stratify women according to their risk of PTB allowing the early identification of mothers who are more likely to have a preterm delivery. The predictive performance proved to be comparable in Northern and Central Italy, while a lower discriminatory power was observed in the Sicily region. This could be partially explained by a misclassification of women with respect to the use of assisted medical conception techniques, which is one of the factors that contributes most to the score. In fact, comparing the group of women who experienced PTB with the one who did not, the proportion of women who used assisted reproduction techniques in Sicily did not differ as much as in the other regions (Lombardy: 16.9% vs. 5.1%; Marche: 14.9% vs. 4.1%; Sicily: 3.9% vs. 2.3%). The lack of a substantial difference between groups, considering that medically assisted conception is a well-known risk factor, could be due to errors in the regional database. This misclassification might have generated a conservative estimate of the PTBS performance in Sicily.

### 4.2. Results in the Context of What Is Known

Although moderate, the discrimination power of PTBS is consistent with that of other previously published predictive models. A recently developed neural network-based algorithm for PTB prediction in nulliparous women showed an AUC of 0.60 during the first trimester of pregnancy [17]. Another study reported an AUC of 0.66 for a predictive model based on a set of well-known risk factors for PTB which, unlike the present study, also included information on ongoing pregnancy and history of preterm births [10]. Furthermore, an independent external validation on a Dutch cohort of five pre-existing prediction models (all considering previous preterm deliveries in addition to other maternal characteristics) reported AUCs ranging from 0.54 to 0.67 [18].

### 4.3. Clinical Implications—The Meaning of the Study

During the variables’ selection process, many predictors were identified. While socio-demographic conditions and the use of assisted medical conception techniques are known risk factors, interpreting the association between PTB and the other factors is more challenging. In fact, these identify the use of certain health services, but no causal relationship with the outcome was investigated. However, some of the selected variables are representative of conditions and risk factors studied extensively in the literature. The intake of pancreatic hormones (i.e., glucagon) and drugs used in diabetes, as well as the exemption for diabetes mellitus, identify people affected by diabetes, a condition that has been shown to be associated with PTB [19,20], while agents acting on the renin–angiotensin system, beta-blocking and calcium channel blockers are drugs commonly used for the treatment of hypertension [21], another strong predictor of preterm delivery [22]. Furthermore, both the transplant recipients and systemic lupus erythematosus exemptions identify conditions known to be associated with a high preterm delivery rate [23,24,25,26]. There were very few women admitted to the hospital diagnosed with chronic renal failure; however, their increased risk of PTB is consistent with what was reported in a meta-analysis [27]. The most frequent component of PTBS is the intake of sex hormones and modulators of the genital system. This includes widely used hormonal contraceptives, some of which were associated with PTB in a previous study [28], as well as a variety of other hormonal drugs, that deserve careful investigation in future studies.

### 4.4. Strengths and Limitations

The present study has several strengths. First, the score was developed by monitoring a large cohort of women in a real-world setting. Second, the score performance was validated under different settings, varying in both temporal and geographical conditions. Third, by using electronic health data, the whole medical history of the mothers was taken into account without limiting the analysis to specific risk factors. Finally, the score can be used in combination with information on previous pregnancies and variables detected during gestation for a better PTB risk assessment.

The study also has a number of potential limitations. First, spontaneous deliveries could not be distinguished from programmed deliveries due to data unreliability. To overcome this limit, and attempt to restrict the evaluation to spontaneous deliveries only, the PTBS was validated in the Lombardy region using more restrictive definitions of preterm delivery (<36 and <32 weeks of gestation). In both cases, the discriminatory power was approximately equal to that observed in the main analysis (data not shown). Second, the analysis was restricted to nulliparous women. Albeit our strict inclusion criteria reduced the potential for confounding by including women in their first experience with the pregnancy, the generalizability of our findings to women with previous births requires extreme caution. However, because the history of preterm delivery is considered one of the most reliable criteria for identifying high-risk pregnancies for women with previous births, our score is a useful tool for women for whom there is greater uncertainty about pregnancy. Third, although we attempted to avoid false-positive signals by excluding variables with a frequency lower than 5 and applying the LASSO method, some high weights were assigned to very rare conditions/procedures in our population (e.g., heart failure, other operations on the rectum and perirectal tissue). Nevertheless, the good validation of the score in a different sample of the Lombardy region, as well as in other Italian regions, corroborates the goodness of the developed tool. Finally, the administrative purpose for which the healthcare utilization databases were instituted limits the completeness and accuracy of the medical information reported. For example, no data on smoking habits, BMI and lifestyle were available, and the severity of the comorbidities that lead women to use the services of the NHS cannot be evaluated.

### 4.5. Conclusions

In summary, a new predictive score, able to stratify nulliparous women according to their risk of preterm delivery, was developed and validated using data routinely collected in Italian healthcare utilization databases. Despite the limited discriminatory power, PTBS can be a useful tool for several healthcare professionals (e.g., general practitioner, obstetrician/gynecologist) to identify women at high risk of PTB early in pregnancy as well as for policymakers to guide health planning.

## Figures and Tables

**Table 1 healthcare-10-01443-t001:** Weight and frequencies, in the training set (Lombardy region), of the 26 variables contributing to the Preterm Birth Score (PTBS), selected applying the LASSO method, via logistic regression.

Variable	Frequency (%)			Weight
	Term Birth	Preterm Birth	Total	
	*n* = 117,480	*n* = 9358	*n* = 126,839	
**Drugs**				
Pancreatic hormones	63 (0.05)	33 (0.35)	96 (0.08)	10
Agents acting on the renin-angiotensin system	803 (0.68)	173 (1.85)	976 (0.77)	5
Sex hormones and modulators of the genital system	22,664 (19.29)	3216 (34.36)	25,880 (20.40)	4
Endocrine therapy	567 (0.48)	156 (1.67)	723 (0.57)	3
Drugs used in diabetes	459 (0.39)	112 (1.20)	571 (0.45)	3
Immunosuppressants	266 (0.23)	53 (0.57)	319 (0.25)	2
Corticosteroids for systemic use	13,200 (11.24)	1341 (14.33)	14,541 (11.46)	1
Beta blocking agents	1034 (0.88)	163 (1.74)	1197 (0.94)	1
Calcium channel blockers	498 (0.42)	98 (1.05)	596 (0.47)	1
**Hospital diagnosis**				
Heart failure	3 (0.00)	5 (0.05)	8 (0.01)	8
Chronic renal failure	2 (0.00)	4 (0.04)	6 (0.00)	4
Diffuse diseases of connective tissue	31 (0.03)	13 (0.14)	44 (0.03)	3
**Inpatient procedures**				
Other operations on rectum and perirectal tissue	7 (0.01)	7 (0.07)	14 (0.01)	12
Diagnostic procedures on liver	23 (0.02)	12 (0.13)	35 (0.03)	4
Lysis of peritoneal adhesions	586 (0.50)	132 (1.41)	718 (0.57)	4
**Exemptions**				
Transplant recipients	13 (0.01)	12 (0.13)	25 (0.02)	8
Diabetes mellitus	303 (0.26)	83 (0.89)	386 (0.30)	4
Systemic lupus erythematosus	97 (0.08)	27 (0.29)	124 (0.10)	4
Affections of the circulatory system	343 (0.29)	56 (0.60)	399 (0.31)	1
Chronic (active) hepatitis	270 (0.23)	51 (0.54)	321 (0.25)	1
**Outpatient services**				
Psychiatry	4324 (3.68)	549 (5.87)	4873 (3.84)	1
General consultation	474 (0.40)	86 (0.92)	560 (0.44)	1
**Socio-demographic conditions**				
Age at conception ≥36 years	22,095 (18.81)	2723 (29.09)	24,818 (19.57)	3
Born abroad	14,014 (11.93)	1300 (13.89)	15,314 (12.07)	1
Low education	20,597 (17.53)	1815 (19.39)	22,412 (17.67)	1
**Use of assisted medical conception techniques**	6005 (5.11)	1587 (16.96)	7592 (5.99)	10

## Data Availability

The data that support the findings of this study are available from the Lombardy region, but restrictions apply to the availability of these data, which were used under license for the current study, and so are not publicly available. Data are however available from the Lombardy region upon reasonable request.

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
