# Peer review of "Development and Validation of a Novel Pre-Pregnancy Score Predictive of Preterm Birth in Nulliparous Women Using Data from Italian Healthcare Utilization Databases"

_healthcare, 2022, doi:10.3390/healthcare10081443_

Round 1
Reviewer 1 Report
In this manuscript, a new predictive score for preterm delivery (Preterm Birth Score, PTBS) was developed in nulliparous women using Italian healthcare utilization databases, by incorporating comprehensive risk factors and related information in patients' medical records.
Although the proposed PTBS shows limited discriminatory power in the experiments, the methods used to develop and validate are solid and the authors explored a comprehensive list of features in EHR, which can be insightful for the future studies on predicting preterm birth.
Author Response
In this manuscript, a new predictive score for preterm delivery (Preterm Birth Score, PTBS) was developed in nulliparous women using Italian healthcare utilization databases, by incorporating comprehensive risk factors and related information in patients' medical records.
Although the proposed PTBS shows limited discriminatory power in the experiments, the methods used to develop and validate are solid and the authors explored a comprehensive list of features in EHR, which can be insightful for the future studies on predicting preterm birth.
We thank the reviewer for his/her appreciation of our work.

Reviewer 2 Report
The abstract should be revised as it does not introduce the area of research and the research question.
The introduction should be rewritten professionally.
Please explain the proposed method in more detail, what is the novelty of the proposed method compared to the state of the art?
Current experiments are weak and a fair comparison with other recent methods is very necessary.
I think it would be good to represent the experiment results in the abstract.
The experiment description section is too rough. The description of data collection should be added.
For the experimental results, it will be good to present a statistical test the comparison the results with other published methods.
This can help support the claim of improved results obtained with the studied selection methods.
Another aspect where paper can be improved is motivation and the reason for the given architecture.
The current approach seems to be more like we have these different types of architectures, let's mix them and present results by training them.
It would be of great interest why a particular model was selected and what a particular part in the framework is helping us to learn.
English can be improved. Proofreading should ensure the appropriate use of grammar, tense, and punctuation.
Longer sentences should be converted into smaller ones.
Author Response
The abstract should be revised as it does not introduce the area of research and the research question.
The introduction should be rewritten professionally.
Please explain the proposed method in more detail, what is the novelty of the proposed method compared to the state of the art?
Current experiments are weak and a fair comparison with other recent methods is very necessary.
I think it would be good to represent the experiment results in the abstract.
The experiment description section is too rough. The description of data collection should be added.
For the experimental results, it will be good to present a statistical test the comparison the results with other published methods.
This can help support the claim of improved results obtained with the studied selection methods.
Another aspect where paper can be improved is motivation and the reason for the given architecture.
The current approach seems to be more like we have these different types of architectures, let's mix them and present results by training them.
It would be of great interest why a particular model was selected and what a particular part in the framework is helping us to learn.
English can be improved. Proofreading should ensure the appropriate use of grammar, tense, and punctuation.
Longer sentences should be converted into smaller ones.
We thank the reviewer for his/her comments. Many factors give the novelty of our project. First, we tried to build a score to identify the risk of experiencing a preterm birth at the beginning of pregnancy using secondary date sources without considering what happens during the course of pregnancy. This is particularly innovative as it would allow clinicians to adopt more intensive antenatal care right from the start of the pregnancy if the woman is classified as at high risk of experiencing a preterm birth. The second strong point is that there is no a priori hypothesis about the predictors of preterm birth; therefore, we have considered the knowable. Precisely for this reason, it is impossible to compare our score with the other scores already published in the literature as we cannot evaluate all the clinical variables that are usually used in the other scores already formulated.
We have reviewed the English language throughout the paper.

Round 2
Reviewer 2 Report
his paper needs to include an analysis of the effectiveness and efficiency between previous results (previous studies) and the outputs described in this paper.
The section proposed approach is very short and poor.
The approach paper has to be improved by adding in more detail how or why this improved result was achieved.
What feature(s) of the enhanced system is responsible? The implemented algorithms and mathematical formula?
Include some words about the approach used to develop the system architecture.
I was also concerned about the number of android applications for training and testing.
Also, is the detected number on its own the best way to judge a system performance
Author Response
His paper needs to include an analysis of the effectiveness and efficiency between previous results (previous studies) and the outputs described in this paper.
We thank the reviewer for his/her comment. However, in section 4.2 of our manuscript, we have already reported a substantial comparison with the previous studies in this field.
The section proposed approach is very short and poor.
We thank the reviewer for his/her comments. However, in section “Score development” we have already described in detail the used approach. Which section are you referring to?
The approach paper has to be improved by adding in more detail how or why this improved result was achieved.
As already reported in the first answer, in section 4.2 of our manuscript, we have already reported a substantial comparison with the previous studies in this field. As you can see, we did not find any better results than what was already published. However, the strength of our approach is the fact that, since our score is based on clinical information retrieved by the healthcare utilisation database, it would be easier for the clinician to calculate it. Moreover, this is the only one that allows to predict the risk of preterm birth at the beginning of delivery since the score uses the clinical history information of the mother.
What feature(s) of the enhanced system is responsible? The implemented algorithms and mathematical formula?
We did not understand what you meant by an enhanced system. We have already explained in detail the model used to select the variables useful to predict the risk of preterm birth (section 2.3). To calculate the score, it is sufficient to add the scores assigned to each variable based on the presence or absence of the variable of interest for the woman in question.
Include some words about the approach used to develop the system architecture. I was also concerned about the number of android applications for training and testing.
We thank the reviewer for his/her comments. For the development of the score, we just used the LASSO model for selecting the most important variable in predicting the risk of preterm birth. The variable selection was made in the training set (which corresponds to 70% of the cohort); then, it was validated with both internal (i.e. the validation set corresponding to the 30% of the cohort, which was not included in the training test) and external validation.
Also, is the detected number on its own the best way to judge a system performance
We focused on the AUC of the score, which assesses both sensitivity and specificity of the system. Furthermore, we also considered the calibration of the score which allowed us to evaluate its reproducibility.
